# Automated Rule Checking for MEP Systems Based on BIM and KBMS

**Xuanfeng Xie [1], Jianliang Zhou [1,*], Xuehai Fu [1], Ruoyi Zhang [1], Hui Zhu [1] and Quanxi Bao [2]**

[1] School of Mechanics and Civil Engineering, China University of Mining and Technology, Xuzhou 221116, China; xuanfeng.xie@outlook.com (X.X.); xuehai.fu@outlook.com (X.F.); zruoyi@outlook.com (R.Z.); 15706298215@163.com (H.Z.)

[2] The Third Construction Co., Ltd. of China Construction Eighth Engineering Division, Xuzhou 221100, China; baoquanxi216@163.com

[*] Correspondence: zhoujianliang@cumt.edu.cn

**Abstract:** Due to the growing complexity of mechanical, electrical, and plumbing (MEP) designs and the rules that govern them, performing rule checks manually has become expensive. However, MEP-based rule checking has not received adequate attention compared to automated rule checking in other domains. Based on Knowledge Management and Building Information Modeling (BIM), an automated rule checking system integrated knowledge base management system (KBMS) for model information expansion, information extraction, system integrity checking, and element spacing checking was developed. MEP rules for automated rule checking were collected, optimized, and stored in the MEP knowledge base. The KBMS facilitates the management of MEP rules in the knowledge base. A Revit plug-in of MEP rule checking system was developed including functions of KBMS, Model Integrity Checking, Elements Space Checking, and Locating the non-compliant element in model view. This study integrated both KBMS and BIM technologies to achieve automated rule checking for MEP. This simplifies the process of rule checking of MEP systems in an automated manner.

**Keywords:** MEP; rule checking; KBMS; integrity checking; space checking

## 1. Introduction

Due to the progress of technology and the improvement of living standards, the demand for building functions has increased. As a result, building designs and codes have become increasingly complex, particularly in the area of mechanical, electrical, and plumbing (MEP).

A number of challenges arise from the complexity of MEP. The MEP subsystems are designed by different engineers. The lack of communication between designers may lead to cross-collisions between pipelines in dense areas of MEP components [1]. Additionally, compared to other common domains, MEP codes are not only numerous and varied but also have a more complex information structure and language representation [2]. Therefore, it is expensive to perform rule checking manually.

Traditionally, MEP systems have been designed in a 2D plan mode. In this mode, information about MEP components, and spatial arrangements of MEP components in a building are shown on a plan. Thus, the interactive relationships between components cannot be clearly illustrated. In addition, there are a large number of design drawings across the different professions, as well as some unavoidable oversights in manual procedures. These circumstances make it difficult to check some non-compliant designs.

Building Information Modeling (BIM) is a digital technology that has rapidly developed in the field of construction engineering in recent years. Compared to traditional CAD, BIM is not only for 3D visualization, but more importantly, BIM brings together the benefits of interoperability, building information integration, and conflict reduction [3].

According to the literature, BIM has shown to be quite useful in MEP applications. Wang et al. (2016) developed a practical BIM framework for integrating MEP layout from the basic design to the construction stage [4]. Cho et al. (2019) proposed a symbol recognition framework that can automatically recognize different types of symbols in the MEP drawings and thus reduce the amount of time required to create BIM models manually [5]. Daszczynski et al. (2022) presented basic construction collisions with algorithms of their solutions and described possible financial benefits associated with the use of the BIM approach in MEP [6]. At present, the application of BIM in MEP systems mostly adopts BIM software to carry out 3D modeling and collision checking of MEP systems [1,7]. A BIM software such as Revit allows the modelers to input all information related to MEP components that have been outlined in the design drawings. However, MEP components are often arranged in a concentrated and overlapping manner, so there is a possibility of component omission [8].

In fact, a wealth of knowledge is required in the design and coordination process of MEP systems [9]. Additionally, MEP system-related standards are complex, involving several professions. Different kinds of buildings have different requirements for MEP systems design. Therefore, it is necessary to conduct knowledge management for MEP rules.

Knowledge management mainly includes the processes of production, sharing, application, and innovation of knowledge [10]. The development of information technology in engineering projects requires an efficient method for managing knowledge resources. BIM-based MEP coordination from the perspective of knowledge management has been explored. For example, Korman et al. (2003) studied and proposed a new BIM-based MEP knowledge framework and reasoning structure, which integrates knowledge related to building design, construction, operation, and maintenance [9]. Wang and Leite (2016) proposed a formal scheme for integrating the MEP coordination process of collision features and associated solutions [11]. Hsu et al. (2020) obtained constructors' clash resolution expertise and developed an AI system employing the techniques of machine learning and heuristic optimization to minimize the clashes of MEP systems [12]. However, few studies have conducted MEP pipeline coordination based on the knowledge base, which is an effective way for knowledge management and has been used in the domain of construction engineering.

The Architectural Engineering Construction (AEC) industry has been greatly affected by the implementation of BIM, and the area of code compliance checking is no exception. In recent years, the application of BIM-based rule checking has been explored. Eastman et al. (2009) conducted systematic research on the application of automated rule checking and proposed a framework for rule checking, summarizing the process in four stages: (1) rule interpretation; (2) building model preparation; (3) rule execution; and (4) rule checking reporting [13]. Since then, many studies have explored rule-checking methods or built rule-checking systems in different domains based on this rule-checking framework. In recent years, many scholars have focused their research on the domain of safety inspection. For example, Hossain and Ahmed (2022) presented a rule-checking system that can automatically identify potential fall hazards [14]. Yuan et al. (2019) integrated BIM and PtD into an approach that can assess safety risks [15]. Some scholars explored the application of BIM-based rule checking in the field of fire safety. Kincelova et al. (2020) developed a BIM-based automated code-checking approach to improve fire safety in tall timber buildings [16]. Malsane et al. (2015) focused on building regulations relating to residential fire safety in England and Wales, and established a building model schema for automated compliance checking [17]. BIM-based automated rule checking also plays a role in the design process. Nawari (2011) presented the Automated Code Conformance Checking (ACCC) framework, which can be seen as a complement and extension of the framework [18] proposed by Eastman et al. (2009) [13]. Nov et al. (2021) developed an automated system to identify clashes of the structural strut in the design process of the deep excavation support structure [19]. Kim and Teizer (2014) explored methods to automate the designing and planning of scaffolding systems [20]. With the development of BIM-based rule checking, semi-automated or fully automated rule checking methods

are explored. Zhou and El-Gohary (2017) made an effort to propose a method for automatic rule extraction and transcription to improve the accuracy and efficiency of the rule checking process [21]. Ghannad et al. (2019) proposed a modular rule checking method combined with a visual programming language (VPL) to represent architectural design rules using LRML patterns [22]. Guo et al. (2021) combined multiple information extraction technologies to automate the entire automated compliance checking process [23].

According to the above review, BIM-based rule checking has been applied in many aspects such as structural design, construction safety and protection, fire escape, building energy saving, etc. In spite of the complexity of MEP systems and the difficulty of manual rule checking, however, MEP-based code checking has not received enough attention. Considering the complexity of MEP systems, it is a feasible and effective way to introduce knowledge management and establish a knowledge base management system (KBMS) on the basis of the knowledge base. It is an improvement of the knowledge base [24]. The KBMS can not only provide data sources for the rule checking system but also provide operations such as querying and editing rules. These operations provide immediate decision support for designers. Additionally, since MEP design is typically completed in a single software [7], compared to third-party automated rule checking, the software secondary development approach makes it more convenient to inspect and correct errors.

To address these gaps, the MEP KBMS was established in this research in which users can query, edit, add, delete, etc., MEP rules. On the basis of the MEP KBMS, a MEP rule checking system implemented in the form of a plug-in was developed to conduct compliance checking on the MEP systems integrity of the BIM model, components' properties, and spacing of the MEP elements. Section 2 presents the framework and methodology of the research, which demonstrates the implementation process of the KBMS and the MEP rule checking system. The detailed implementation process of the KBMS is presented in Section 3. The implementation of the MEP rule checking system is presented in Section 4. Three case studies are illustrated in Section 5. A summary of the contributions and limitations is in the discussion section. The paper concludes with suggested directions for future research.

## 2. Methodology

### 2.1. Research Framework

The objective of this study is to realize automated rule checking for MEP by integrating both MEP KBMS and BIM-based MEP rule checking system. Figure 1 illustrates the overall framework created in this study, which consists of the following four steps.

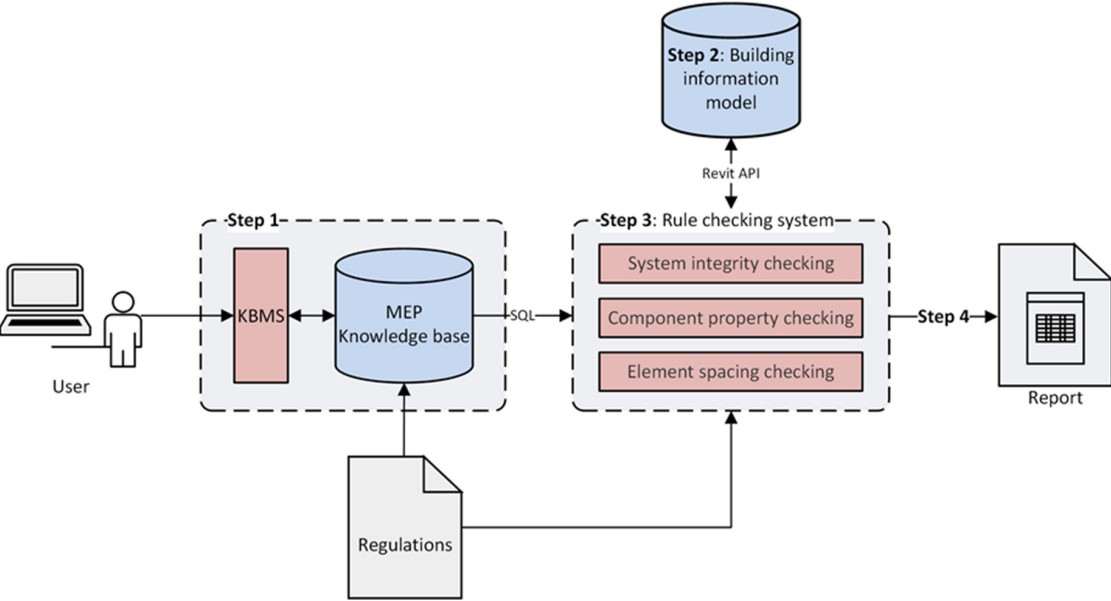

**Figure 1.** Framework of this study.

1.  Create the MEP KBMS. To begin with, suitable rules (discussed in Section 3.1) were extracted from both MEP regulations and BIM delivery standards. After classification, the production representation was used to convert the rules from unstructured language into structured language, which could then be conveniently stored in the knowledge base. To facilitate user operation, the management system was established based on the knowledge base.

2.  Establishment of a BIM model using Revit (BIM software developed by Autodesk$^{TM}$). For compliance checking, the BIM model provides primary data information.

3.  Secondary development of Revit to develop MEP rule checking system. The rule checking system consists of three modules: system integrity checking, component property checking, and element spacing checking. The checking system is equipped with two types of data extraction: the extraction of information from the extended BIM model using API and the extraction of information from the Knowledge base of rule about MEP using SQL language. Not only were the rules employed to generate the knowledge base, but they were also converted to logical expressions (in the form of C# programming language) that computers are able to handle, and the extracted data were then fed into the corresponding logical expressions for analysis.

4.  Visualization of the checking results. In addition to a text-based report, the results will also be presented in an interactive format. The text-based checking results include the name of the non-compliant components and the basis for checking, and the component in error will be automatically located and highlighted in the view after the user clicks the corresponding button.

This study proceeds around the implementation of MEP KBMS and the MEP rule checking system.

### 2.2. Process of Establishing the MEP KBMS

The process of establishing the MEP KBMS is as follows: collecting the MEP knowledge, optimizing the MEP knowledge, establishing the MEP knowledge base, and establishing the MEP KBMS. Its basic composition is shown in Figure 2. Among them, the fact base is used to store all kinds of factual knowledge, such as the parameters of MEP components and each MEP professional type, etc. In the rule base, the knowledge representation method was used to store the inspirational knowledge of MEP pipeline synthesis, i.e., the collection of all kinds of rules used to solve the problem, which expresses the logical relationships among the various types of knowledge in the knowledge base. Section 3 describes the implementation process of the KBMS in detail.

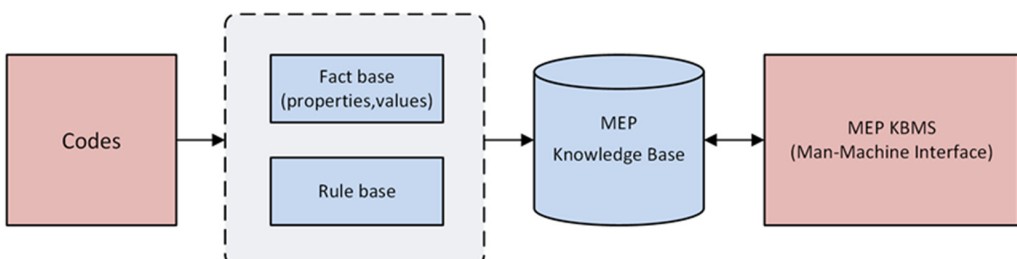

**Figure 2.** Creation process of the MEP KBMS.

### 2.3. Process of Establishing the Rule Checking System

Secondary development was used to implement the rule checking system on the BIM platform. By comparison with third-party compliance checking software based on IFC models, plugins based on BIM platforms facilitate the inspection process. As well as preventing data compatibility issues, plugins may also assist designers in identifying designs that do not comply with the rules during the design process without having to modify them afterward [15]. In this study, a Revit plugin was developed to automate the checking of rules.

Figure 3 illustrates the implementation steps and architecture of the rule checking system. The checking system is separated into three logical tiers: Data tier, logic tier, and Presentation tier. On the basis of the MEP knowledge base and BIM models created through Revit, the data tier of the checking system can be fully realized through Step 1 (Model information extraction). The logic tier consists of Step 2 (Model information extraction) and Step 3 (Rule execution). Step 4 (Rule check reporting) is presented to users via the Presentation tier of the checking system. Among them, rule execution can be divided into three parts according to the three types of rule checking. The MEP rule checking system is explained in more detail in Section 4.

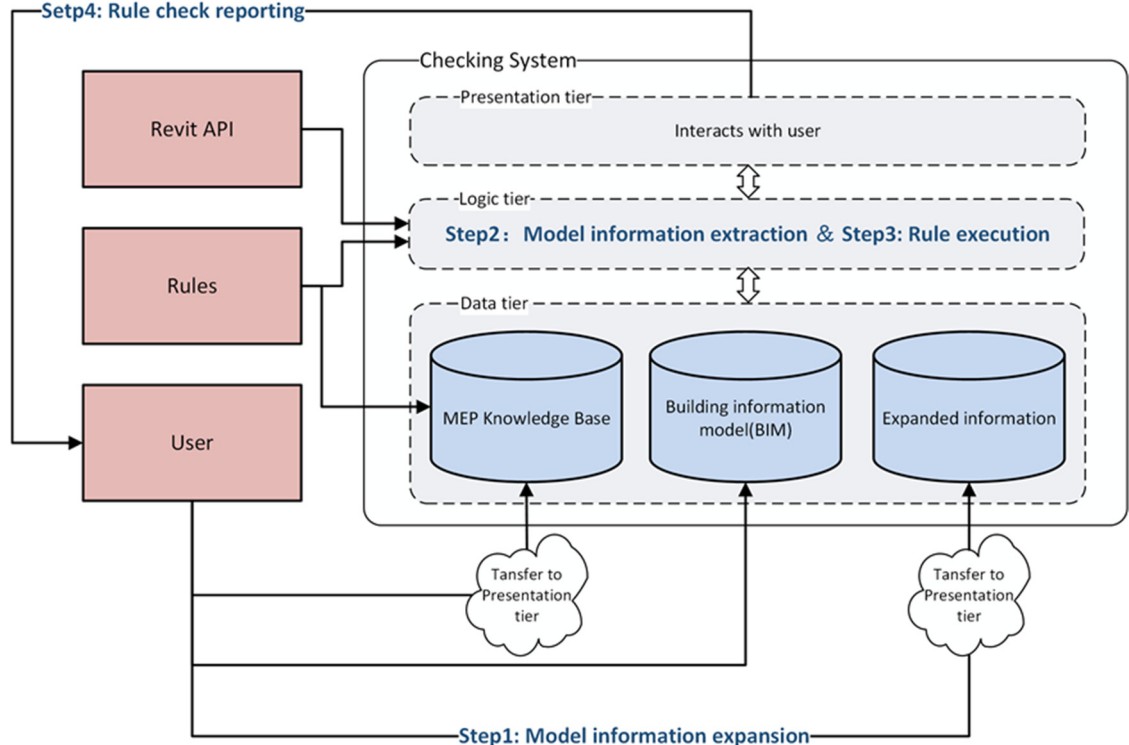

**Figure 3.** Architecture and implementation of the rule checking system.

## 3. Implementation of the MEP KBMS

The objective of MEP KBMS is to store MEP-related codes and regulations to encourage the learning, sharing, and innovation of information. The MEP KBMS implementation procedure is described in this section.

### 3.1. Collecting the MEP Knowledge

MEP systems involve a number of codes, which can be classified into two general categories in this study: (1) rule of design and installation and (2) rule of BIM delivery [25].

This study collected rules suitable for automated inspection from a large number of MEP design and installation codes. As described by Solihin and Eastman [26], the rules are divided into the following types:

1. Rules that require a single or small number of explicit data;
2. Rules that require simple derived attribute values;
3. Rules that require extended data structure;
4. Rules that require a "proof of solution".

This study focuses on the rules of type 2 and 3. Following that, two types of rules were filtered out, which included rules of properties and rules of spacing constraints between MEP components.

For BIM delivery requirements, it adheres to BIM delivery standards, which require BIM model accuracy, content, and naming standards. For instance, according to China's Standard for design delivery of building information modeling (GB/T51301-2018), MEP systems classification should conform to the provisions of Table 1. These codes mainly address the systematic classification and naming of components. According to the needs of front-line designers, an automated MEP systems integrity check of the BIM model was implemented in this study.

**Table 1.** MEP systems classification in China (part).

| Level 1 System | Level 2 System | Level 3 System |
|---|---|---|
| Water Supply and Drainage System | Water Supply System | Water Supply System<br>Hot Water System<br>Direct Drinking Water System |
| | Drainage System | Sewage and Waste Water System<br>Rainwater System |
| | Reclaimed Water System | Reclaimed Water Treatment System<br>Reclaimed Water Supply System |
| | Circulating Water System | Cooling Circulating Water System<br>Swimming Pool Circulating Water System<br>Waterscape Circulating Water System |

### 3.2. Optimizing the MEP Knowledge

The expression format of rules is not fixed, and their textual information is expressed in the form of unstructured language [27]. Therefore, it is necessary to transform the unstructured language into structured language by applying knowledge representation [28].

Knowledge representation mainly consists of predicate logic representation, production rule representation, framework representation, semantic network, object-oriented representation, etc. [29] (pp. 16–25). The production rule representation [30], which is expressed in the form of "IF P THEN Q", is basically consistent with the structure of the filtered rules. In the formula, P indicates whether the formula is valid, while Q represents the conclusion or operation after P is established. For example, "the general slope of the building plastic branch drain is 0.026" can be expressed as IF the material of the building branch drain is plastic, THEN the pipe's general slope is 0.026.

P and Q include one or more sequences of words, each of which is determined by "Component", "Property", "Comparative Words", and "Value". A sequence of words constitutes a physical attribute, and the linear combination of such "physical property" represents rules, as shown in Figure 4. It can be used as the minimum structure for rule checking.

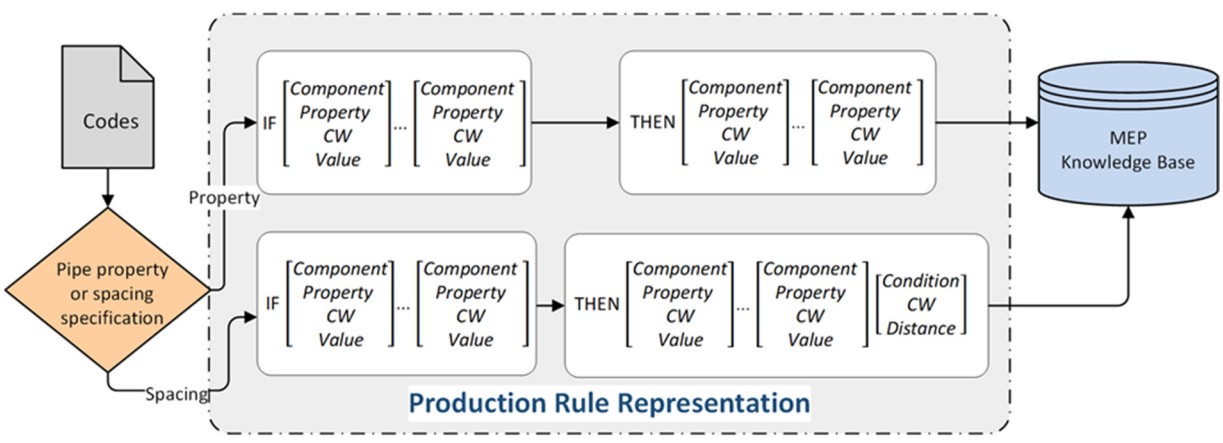

**Figure 4.** Process of optimizing the MEP knowledge.

"Component" is both the object of a code constraint and the subject of rule checking. The value of "Property" could be slope, material, pipe diameter, etc. The "Comparative Word (CW)" refers to a logical relation such as "is", "not greater than", "less than", etc. The "Value" is a numerical type, i.e., a quantifiable value.

For the rules of the pipeline's own properties, multi-dimensional "physical properties" can be used to represent. For example, "if the diameter of the plastic building drainage horizontal main pipe is 110 mm, the general slope is 0.012" can be expressed as the following Table 2.

**Table 2.** The example of optimizing the rule about pipeline's own properties.

|  | Component | Property | CW | Value |
|---|---|---|---|---|
| **IF** | Building drainage horizontal main pipe | Material | is | Plastic |
| **AND** | Building drainage horizontal main pipe | Diameter | is | 110 mm |
| **THEN** | Building drainage horizontal main pipe | General slope | is | 0.012 |

And for the spacing constraints between the pipelines, rules can be expressed by combining physical properties with an additional dimension, which includes "Condition", "Comparative Words", and "Distance". Among the options for "Condition" are empty, cross-laying, parallel laying, etc.

For example, "when the vertical pipe's diameter is not more than 32 mm, the net distance to the wall is not less than 25 mm" can be expressed in accordance with the following Table 3.

**Table 3.** An example of optimizing the component spacing rule.

| **IF** | **Physical Property** | **Component** | **Property** | **CW** | **Value** |
|---|---|---|---|---|---|
|  |  | Vertical Pipe | Diameter | Not Greater Than | 32 mm |
| **AND** | **Physical Property** | Wall | NULL | NULL | NULL |
| **AND** |  | **Condition** |  | NULL |  |
| **THEN** |  | **CW** |  | Not greater than |  |
|  |  | **Distance** |  | 25 mm |  |

Another example is "when the horizontal main pipe is laid in parallel, the horizontal net distance with the drainage pipe is not less than 200 mm", which can be expressed in accordance with the following Table 4.

**Table 4.** An example of optimizing the component spacing rule including laying condition.

| **IF** | **Physical Property** | **Component** | **Property** | **CW** | **Value** |
|---|---|---|---|---|---|
|  |  | Horizontal Main Pipe | NULL | NULL | NULL |
| **AND** | **Physical Property** | Drainage Pipe | NULL | NULL | NULL |
| **AND** |  | **Condition** |  | Parallel laying |  |
| **THEN** |  | **CW** |  | Not less than |  |
|  |  | **Distance** |  | 200 mm |  |

### 3.3. Establishing the MEP Knowledge Base

After determining the knowledge specification expressions, the MEP knowledge base was created by using the relational database Microsoft Access. Knowledge query and reasoning were realized by designing the tables, fields, and inter-table relationships of the database, and seven fact tables and spacing constraints rules table were built to complete the design of the storage structure of the knowledge base.

Primary key and foreign key constraints define the one-to-many and many-to-many linkages between tables in the MEP knowledge base, and the relationships between these tables are described (as shown in Figure 5), forming the knowledge base's overall network architecture.

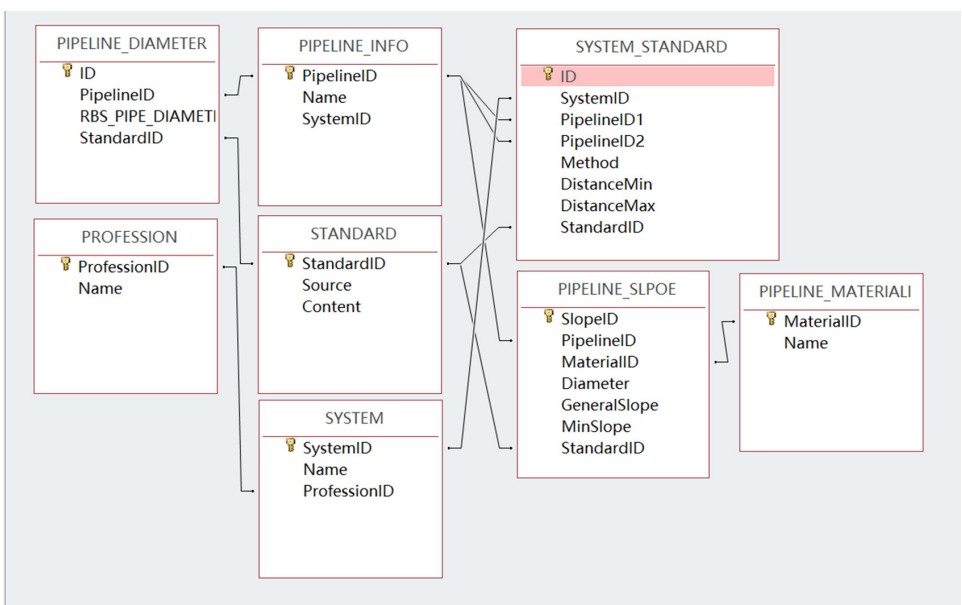

**Figure 5.** Relationship among the tables of the MEP knowledge base.

### 3.4. Establishing the MEP KBMS

The Knowledge base of rules about MEP established above provides storage for knowledge expression. However, a database platform (Microsoft Access) is required for its use, which is not convenient for ordinary users to perform operations such as updates and queries, so the MEP KBMS is set up.

The process is as follows.

1. Create a window through the C# programming language and the use of WPF (Windows Presentation Foundation) framework, which can separate the work of interface designers and developers.
2. In the .cs file, introduce the namespace System.Data, System.Data.OleDb, pass in the database-related parameters to create OleDbConnection to connect to the Access database, then use connection.Open() to open the database connection.
3. In the .xaml file, adjust the shape of each component size, location, and direction to complete the UI design. Through the data binding to obtain user input data, operations, and output execution results.
4. After processing user input, execute SQL statements through the CommandText function in the class named OleDbCommand.
5. Create an object through OleDbDataReader, which is the result of the query saved by the ExecuteReader function in the OleDbCommand class, and then display the required information on the form by reading it out in a loop.

Through different operations of SQL statements, such as SELECT, DELETE, UPDATE, and INSERT, the MEP KBMS, which is shown in Figure 6, implements the functions of adding, deleting, finding, modifying, resetting, etc.

| Select | ID | Pipe Name | Diameter | Material | Regular Slope | Minimum Slo |
|--------|-----|-----------|----------|----------|---------------|-------------|
| ☐ | 23 | Horizontal main drain | 110 | Plastic | 0.005 | 0.003 |
| ☐ | 24 | Horizontal main drain | 125 | Plastic | 0.012 | 0.004 |
| ☐ | 25 | Horizontal main drain | 160 | Plastic | 0.01 | 0.0035 |
| ☐ | 26 | Horizontal main drain | 200 | Plastic | 0.007 | 0.003 |
| ☐ | 27 | Horizontal main drain | 250 | Plastic | 0.005 | 0.003 |
| ☐ | 28 | Horizontal main drain | 315 | Plastic | 0.005 | 0.003 |
| ☐ | 17 | Sanitary waste pipe | 50 | Cast iron | 0.035 | 0.025 |
| ☐ | 18 | Sanitary waste pipe | 75 | Cast iron | 0.025 | 0.015 |
| ☐ | 19 | Sanitary waste pipe | 100 | Cast iron | 0.02 | 0.012 |
| ☐ | 20 | Sanitary waste pipe | 125 | Cast iron | 0.015 | 0.01 |
| ☐ | 21 | Sanitary waste pipe | 150 | Cast iron | 0.01 | 0.007 |
| ☐ | 22 | Sanitary waste pipe | 200 | Cast iron | 0.008 | 0.005 |
| ☐ | 31 | Gravity flow drains suspension pipe | 110 | Plastic | | 0.005 |
| ☐ | 31 | Gravity flow drains suspension pipe | 100 | Cast iron | | 0.01 |
| ☐ | 33 | Rainwater drainage pipe | 110 | Plastic | | 0.005 |
| ☐ | 33 | Rainwater drainage pipe | 100 | Cast iron | | 0.01 |
| ☐ | 34 | Household pipe for outdoor domestic drainage of community | 160 | Cast iron | | 0.005 |
| ☐ | 35 | Outdoor domestic drainage branch pipe of community | 160 | Cast iron | | 0.005 |
| ☐ | 36 | Outdoor domestic drainage main pipe of community | 200 | Cast iron | | 0.004 |
| ☐ | 36 | Outdoor domestic drainage main pipe of community | 315 | Cast iron | | 0.003 |
| ☐ | 12 | Inlet pipe | 20 | | | |

**Figure 6.** The MEP KBMS window.

## 4. Implementation of the Rule Checking System

To bridge the gaps between the MEP KBMS and BIM, this study developed a rule-checking system that can identify components or areas in models that do not comply with the requirements of the rule. The following section describes the implementation process in detail.

### 4.1. Model Information Expansion

In contrast to 2D CAD, which can only add textual information to geometry and its layers [31], BIM introduces an object-oriented data modeling methodology [32]. Consequently, BIM model components typically contain types and a variety of corresponding attributes, which makes rule checking considerably more convenient.

In spite of the fact that the model information contained in BIM is useful, it is insufficient to fully meet the requirements of rule checking. Therefore, it is necessary to expand the model information.

This study expands the Revit model information to ensure the accurate extraction of inspection objects. Parameters of components in the Revit model are based on physical attributes. In addition, "Standard for graphic expression of building information modeling" (JGJ/T 448-2018) also requires pipes to be named in the form of professional abbreviation-pipe type-nominal diameter, such as P-water supply-DN100. However, the descriptions of components in the standard are primarily dependent on their functions, such as the inlet pipe and the horizontal branch drain. Due to the fact that models in Revit rarely provide information regarding the functions of components, and it is challenging to infer the functions of components from their other properties and spatial information, it is nearly impossible to identify the components to be inspected solely from the original information associated with the Revit model. To solve this problem, this study introduces information expansion methods to label or supplement the corresponding components. The following two labeling methods have been developed based on user behavior.

- Label separately. Having selected a single component, the user can choose the corresponding function-based component type in the properties panel to complete the marking process. The panel is shown in Figure 7.

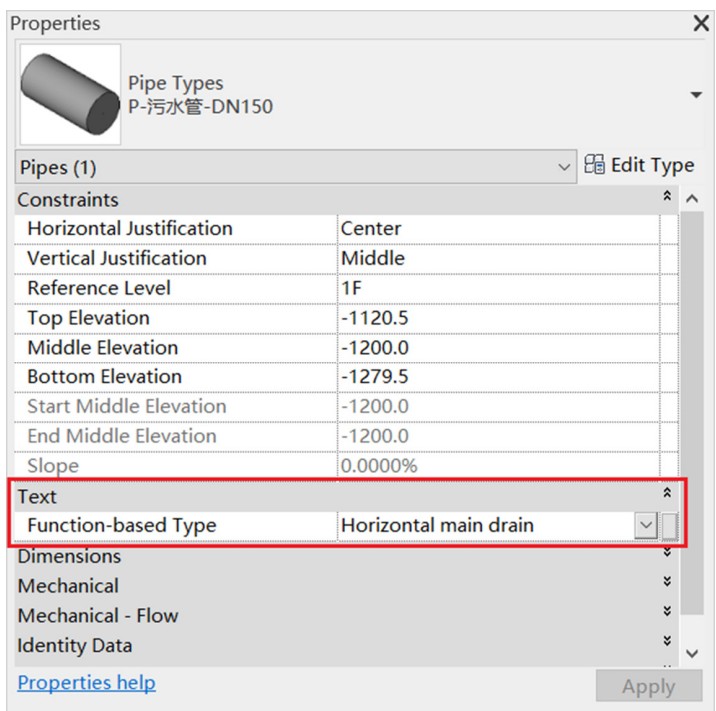

**Figure 7.** The properties panel for labeling separately.

- Label centrally. Once users select a specific function-based component type in a window, Revit enters the state of pending selection automatically, which allows users to mark multiple components of the same type at the same time. The window is presented in Section 5.2.

Regardless of the labeling method employed, the labeling information is stored in the shared parameters created through the API. As a result, even if the plug-in or Revit application is closed, the labeling information is preserved.

### 4.2. Model Information Extraction

The extraction of model information can be conducted once BIM models are sufficiently complete and the model information has been expanded by the checking system. As illustrated in Figure 8, the steps of the extraction algorithm are separated into two categories: object extraction and parameter extraction.

The process of extracting objects before extracting component parameters is an essential step, whether it is system integrity checking, component property checking, or element spacing checking. In Revit, the majority of objects that are visible to users are elements, such as columns, families, family types, and family instances. The following three approaches are used to extract objects according to various usage scenarios.

- Obtain elements by ID. Every Element in Revit is uniquely identified by an ID. An element can be obtained by calling the corresponding method with the ID as a parameter. However, since this method requires knowing the ID in advance, its applications are limited.
- Obtain elements by user selection. There are two ways to perform this method: selecting the element first and then executing the program, and executing the program first and then selecting the element. The following example illustrates the latter approach. Initially, the selection range should be restricted by a filter, after which the

user may only pick the elements that can pass the filter. Once the PickObjects method is called with the filter as an argument, Revit will automatically enter the pending selection state, enabling the user to select one or multiple elements. In the end, the elements selected by the user will be stored in a collection for further use.

- Obtain elements through ElementFilters. A combination of a collector and filters is used to implement this method. The collector is a collection of elements, and the filter is used to set filtration conditions. There are three steps in the implementation process. To begin, an instance of the FilteredElementCollector class is created. In the next step, filters are created according to the filter criteria. Finally, apply one or more filters to the FilteredElementCollector instance to retrieve the elements that are eligible. Due to the large number of built-in filters available in the Revit API, this approach is highly flexible and widely used.

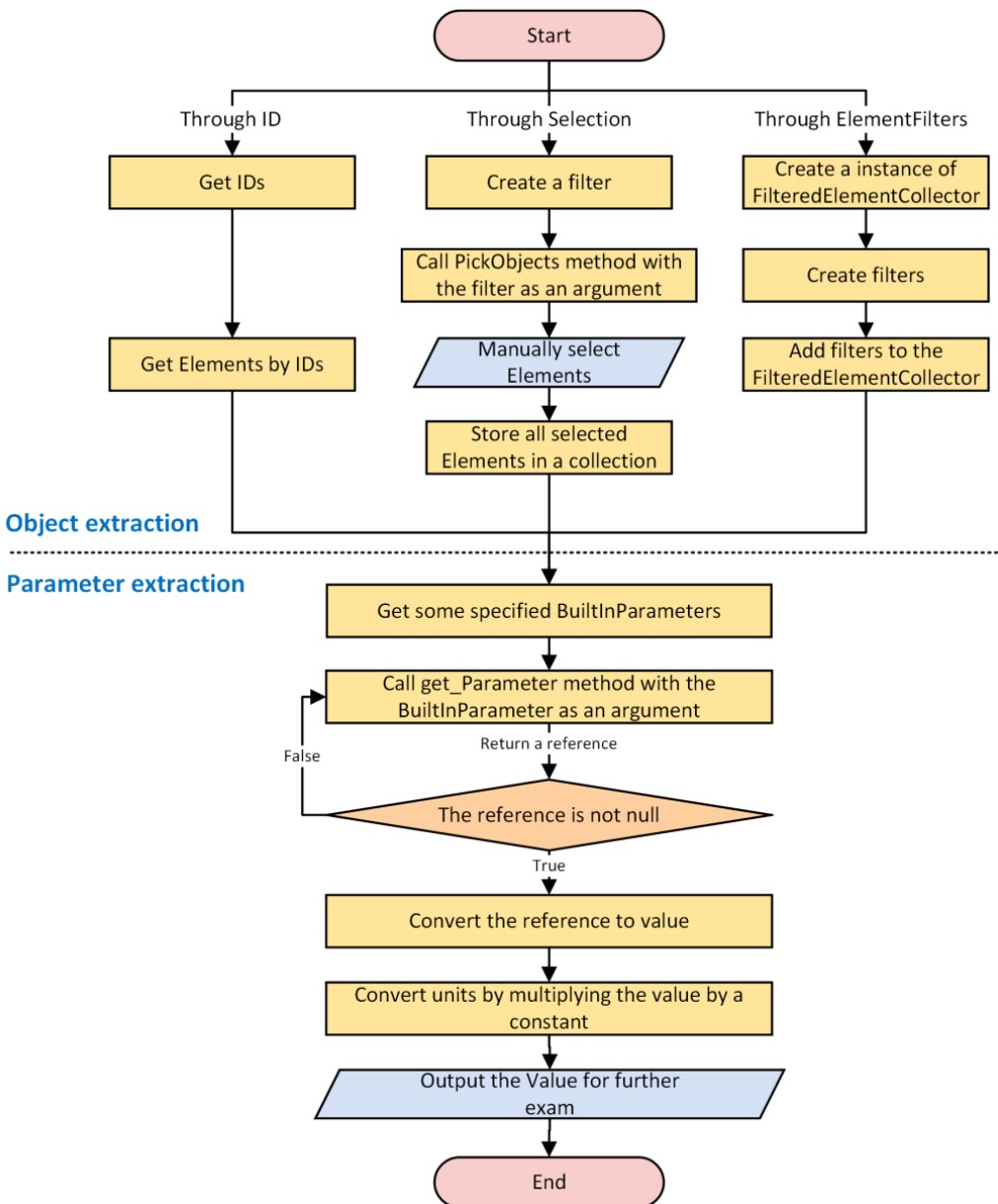

**Figure 8.** Algorithm for model information extraction.

As soon as the necessary check object is extracted, its parameters can be obtained readily. The specific operation process is divided into 5 steps.

1.  Obtain the BuiltInParameter that corresponds to the required parameter; this can be carried out by using the Revit Lookup tool included in the Revit SDK.
2.  Obtain the reference type of the parameter by calling the get_Parameter method of the selected element with the acquired BuiltInParameter as its argument.
3.  Determine whether the reference is empty since not all Elements contain corresponding parameters.
4.  The reference type will be converted to a value type if the reference is not null.
5.  To convert units, multiply the value type of the Element parameter by a constant. There may be differences in the units of measurement used by different countries. As an example, in the building standard "Standard for the distribution of water supply and drainage" (GB50015-2019), the measurement unit of pipe diameter is millimeters, while length is measured in feet in Revit. Consequently, after the diameter of the pipe has been extracted, it is necessary to multiply it by a constant to convert the unit.

*4.3. Rule Execution*

The rule execution process can be carried out by utilizing information extracted from both the model and the previous MEP knowledge base. The implementation of the rule execution phase is shown in Figure 9. Initially, the rules are converted into the internal form of the inference engine (which is implemented in C# in this study). The previously extracted data are then delivered to the inference engine, which then queries the MEP knowledge base for information if necessary. Finally, a logical judgment is made and the non-compliance set is obtained. In addition, data validity and data consistency should be assessed before formal checking. In this section, three checking modules are further presented: (1) System integrity checking, (2) Component property checking, and (3) Element spacing checking.

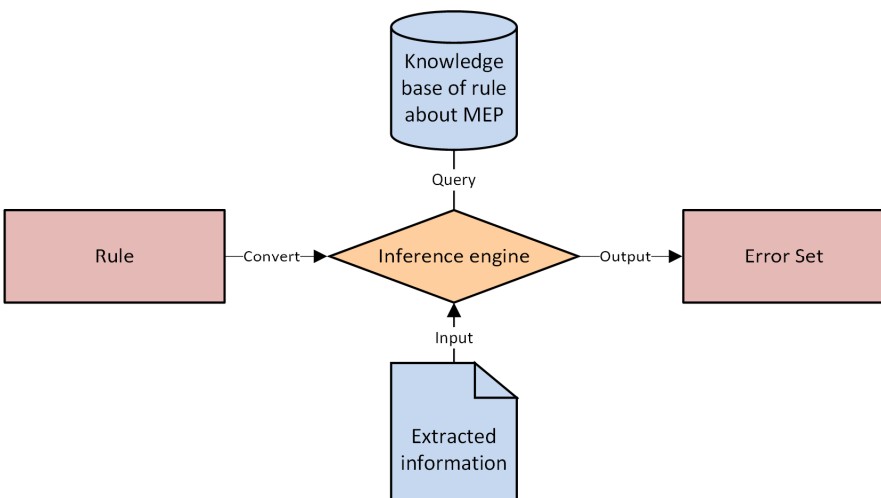

**Figure 9.** Implementation of the rule execution phase.

4.3.1. System Integrity Checking Module

To ensure the MEP systems integrity of the BIM model, each component's connection relationships must be assessed. In Revit, each end of a MEP component contains a Connector. As an example, every pipe and elbow contain two Connectors, while every T-junction contains three Connectors. It should be noted that when the Connector is connected, the connection will also include Connectors of other components, and the Connector is then able to identify not only its Owner (host), but also other Connectors connected to the connection. Based on the characteristics of Connectors, recursion and looping can be used to obtain all the components connected to a component. Figure 10 illustrates the integrity checking algorithm.

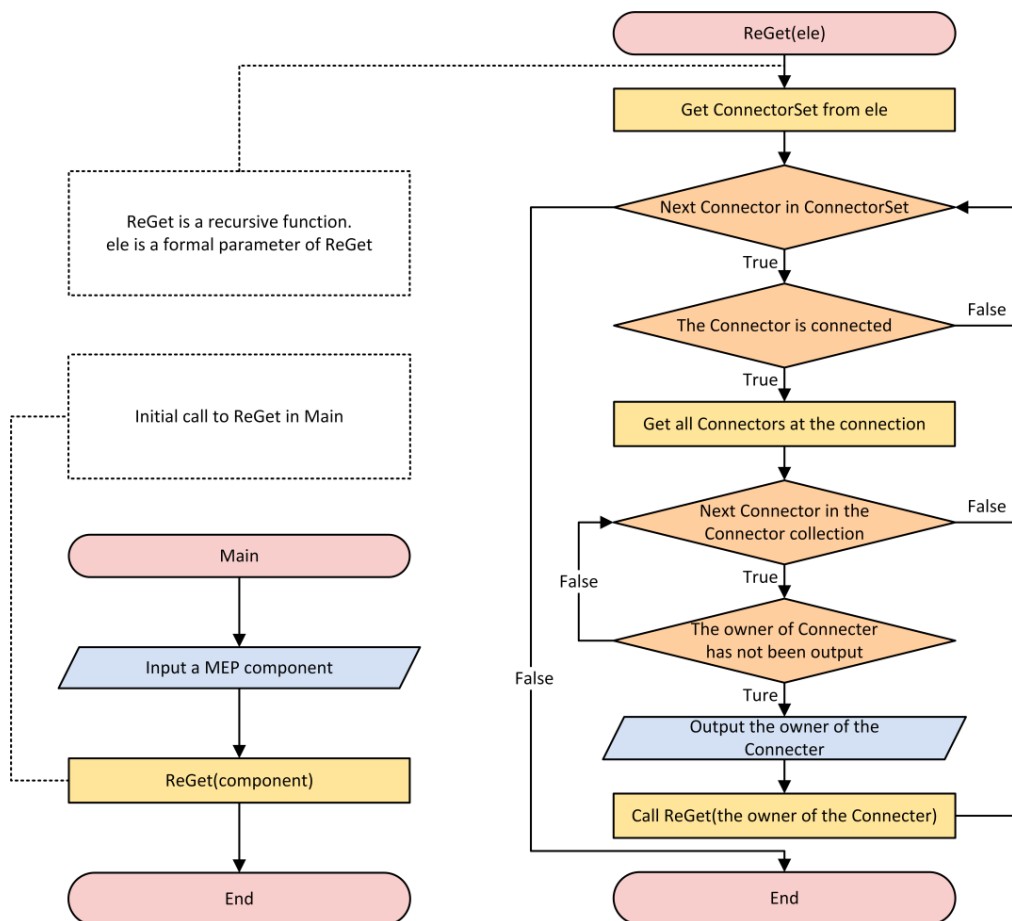

**Figure 10.** Algorithm for system integrity checking.

### 4.3.2. Component Property Checking Module

For component property checking, the task of the inference engine is comparing the extracted information with the information in the MEP knowledge base. Property checking is performed on individual components, so the algorithm used for checking one component versus hundreds of components does not differ significantly. One loop constitutes the body of the checking procedure, and therefore its running time increases linearly with input size, making it suitable even for large input volumes.

### 4.3.3. Element Spacing Checking Module

Element spacing checking, as compared with component property checking, is characterized by a much more complex inference engine, and its primary content of checking is the spatial relationship between elements.

The spacing checking algorithm is implemented in five steps, as illustrated in Figure 11. (1) Query whether there is a spacing requirement for the two elements in the knowledge base. (2) If the spacing requirement exists, obtain the actual spatial relationship between the two elements, such as whether they are laid in parallel or crossed. The analysis of the spatial relationship is based on the correspondence between the vectors contained within the elements. (3) Query the spacing value required by the rule through the two members and their spatial relationship. (4) Calculate the actual spacing of the two elements, which is achieved by the position coordinates of the elements. (5) Compare the actual spacing with the spacing required by the rule.

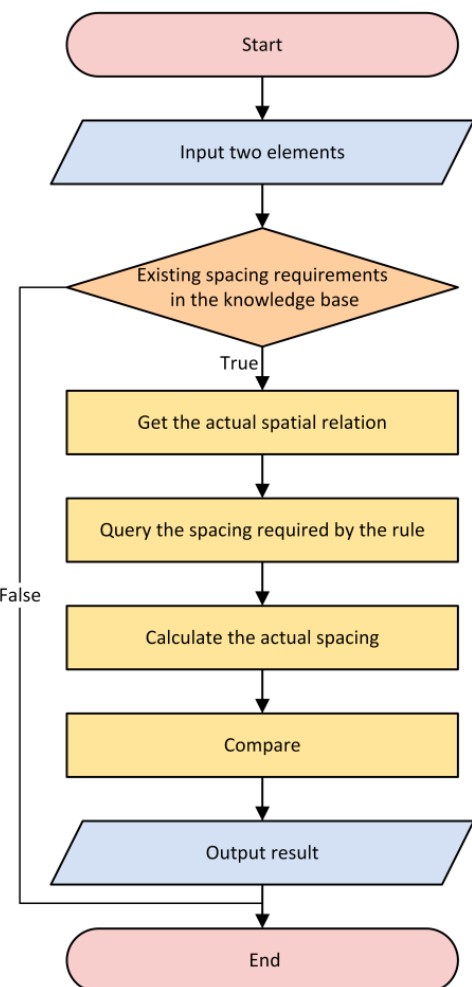

**Figure 11.** Algorithm for element spacing checking.

*4.4. Rule Check Reporting*

To make it easier for users to view and correct non-conformities in the design as early as possible, rule-checking results are presented in two forms: (1) Pop-up alert windows containing the names of the wrong components and the basis of checking. (2) Highlighting the non-compliant components or areas in the model view, which is triggered by clicking the wrong component button in the alert window. More information regarding both forms is presented in Section 5.

## 5. Case Studies

Three case studies were performed to validate the rule checking system based on the BIM platform. A plumbing model of an eight-story comprehensive office building conforming to the codes was created through Revit. On the basis of the compliant model, 15 integrity errors, 15 slope-related property errors, and 10 element spacing errors were manually set to evaluate the checking system, as shown in Figure 12.

The rule checking system was created by integrating several different modules on the basis of an external application. Having compiled the source code into a .dll file, a .addin file is required to be generated in the directory specified by Revit for registration. Upon starting, Revit will automatically locate the addin file in this directory and launch the rule checking system. After the loading is complete, a new MEP Check tab will be added to the Ribbon above Revit, as illustrated at the top of Figure 12.

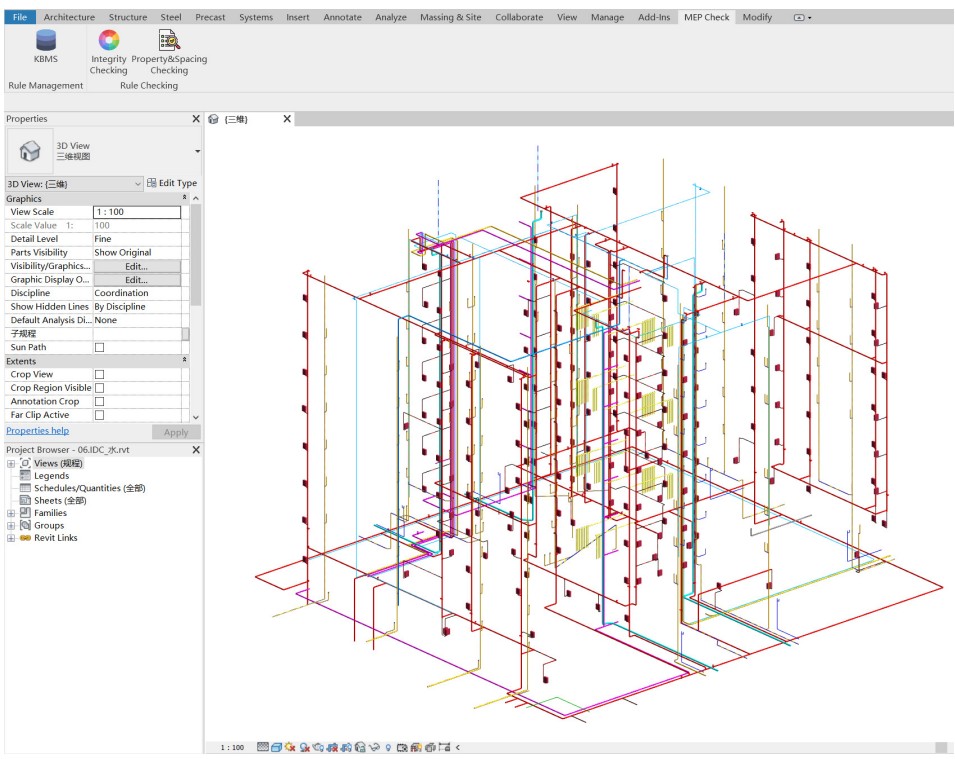

**Figure 12.** The BIM model and the MEP check tab.

### 5.1. Case Study: System Integrity Checking

"Standard for design delivery of building information modeling" imposes numerous requirements on BIM models, and this section describes the process of checking system integrity using the checking system.

Click the "Integrity Checking" icon on the tab to enter the user interface of MEP Systems integrity checking. The sub-tabs of the window are clearly visible, including three tabs about plumbing checking, mechanical checking, and electrical checking, as shown in Figure 13.

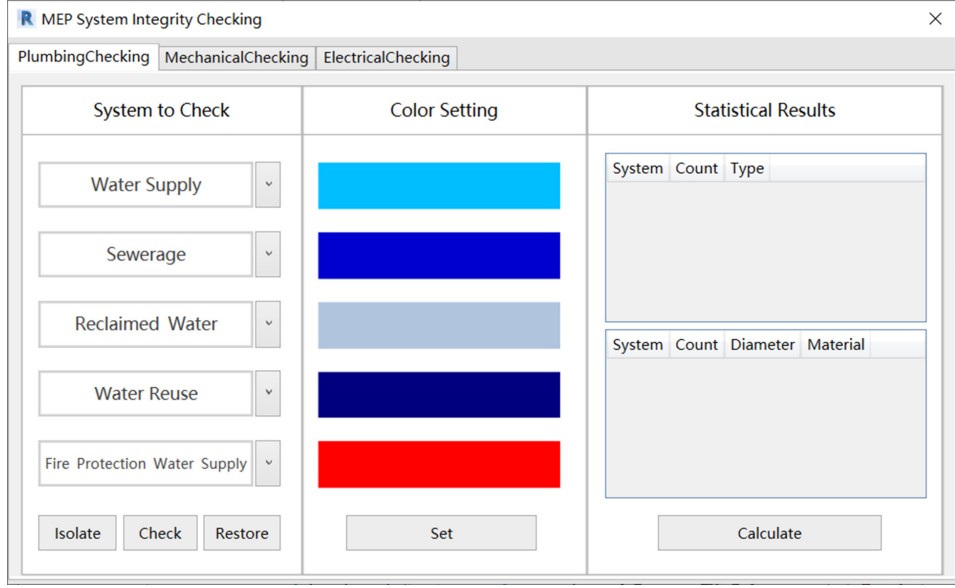

**Figure 13.** The user interface of MEP Systems integrity checking.

After clicking the "Set" button, the color of the model will be set according to the code.

Choose the indoor fire hydrant system within the fire protection water supply system and then press the "Isolate" button. The system will be fully selected and temporarily isolated in the view.

Following completing the operation of isolating the chosen system, click on "Check", and the system will execute the system integrity check algorithm mentioned in Section 4. Figure 14 illustrates the result of the system integrity check. The view will automatically gather the devices in the selected system that are not connected to the related pipeline. According to the results, all 15 manually set errors have been detected. This is convenient for users to quickly locate the unconnected devices to check for omissions.

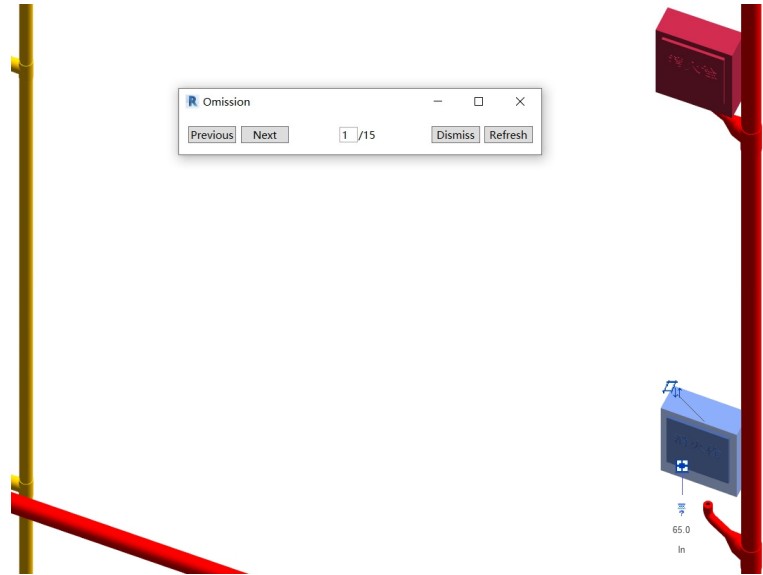

**Figure 14.** Checking results of the system integrity.

*5.2. Case Study: Component Property Checking*

The most common rule class is property definition. This section demonstrates the process of checking pipe slopes. By clicking on the "Property and Spacing Checking" icon on the tab, the user interface of MEP Property and Spacing Checking will be displayed as shown in Figure 15.

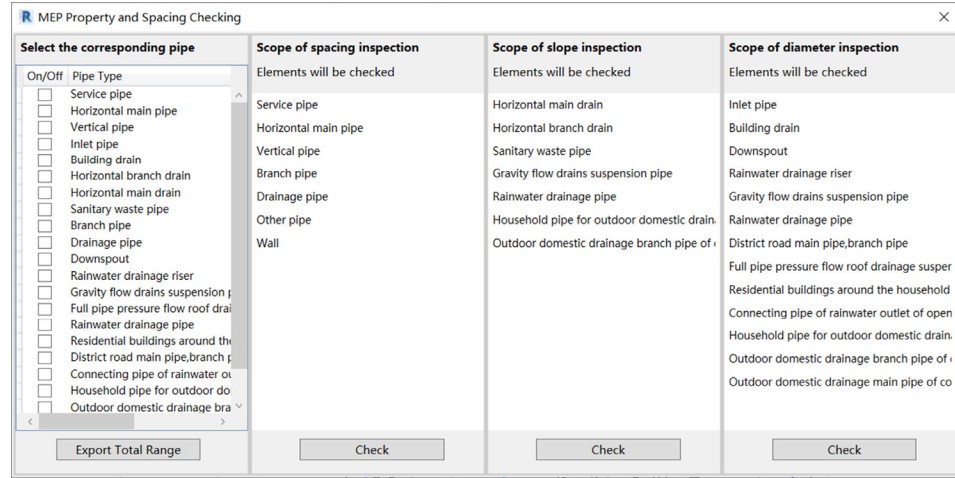

**Figure 15.** The user interface of property checking and spacing checking.

The first column illustrates the pipe type that can be labeled. To expand the information, either put a tick in the check box to label centrally, or label separately as indicated

before. The types of pipes that could be inspected for slope are shown in the third column of the window. This information will be extracted from the knowledge base whenever the window is opened. Clicking on the second "Check" button will trigger the system to assess the slope of the pipes using the algorithm described in Section 4, and the result is visually presented in Figure 16 The results indicate that all 15 manually set errors have been detected. By clicking the button containing the name of the pipe type, the view will be automatically positioned to the non-compliant pipe and highlighted. In the check note, it is specified that when the building's horizontal main drain is made of plastic and the pipe diameter is 110 mm, the slope must be at least 0.0040. According to the property window, the slope of the non-compliant pipe in the figure is zero, which does not comply with the code requirement.

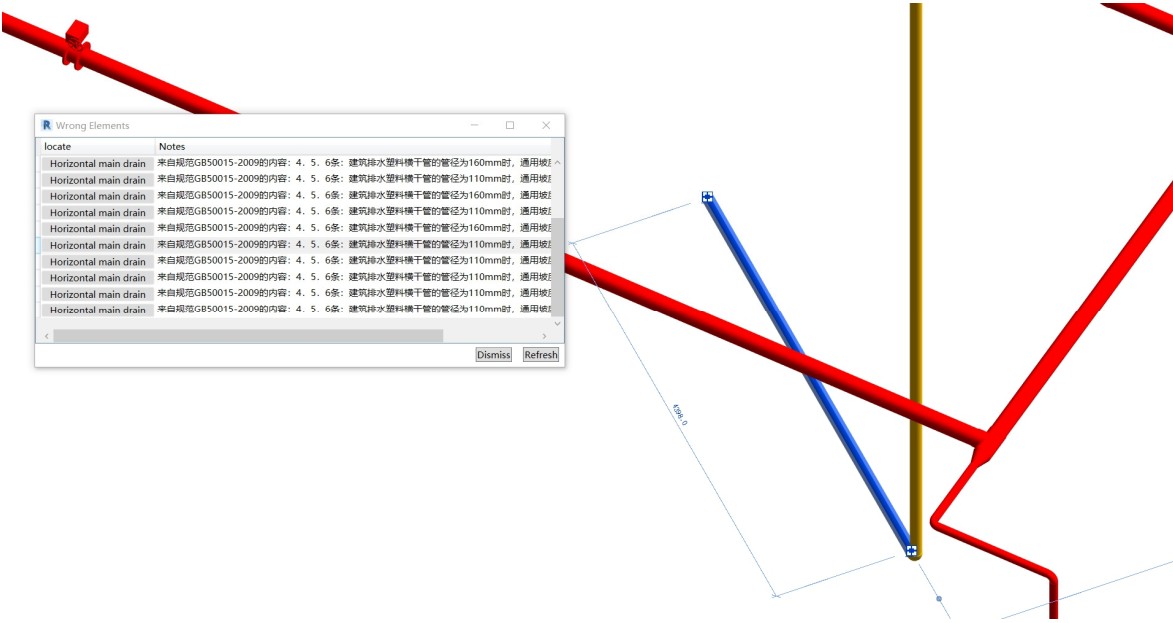

**Figure 16.** Checking results of the component property.

*5.3. Case Study: Element Spacing Checking*

A large number of rules are included in the codes concerning spacing requirements, which are laborious to review manually. The following section describes the process of checking pipe spacing.

The spacing checking module shares the window with the property checking module. The second column in the figure shows the types of pipes that could be inspected for spacing. This content is also pulled from the knowledge base when the window is launched. By clicking the first "Check" button, the system executes the algorithm for spacing check in Section 4, and the results are also displayed to the user in a window. By clicking on the button that holds the name of the pipe type in the window, the view is automatically positioned to the pair of elements that do not meet the requirements of the codes. In accordance with the note in Figure 17 the distance between the main horizontal pipe and the other pipes should not be less than 100 mm. The distance between the two pipes in the figure is less than 100, which does not match the rule.

The results indicate that all manually set errors have been detected. However, this window contains 14 error messages due to the fact that the same group of elements may violate multiple codes.

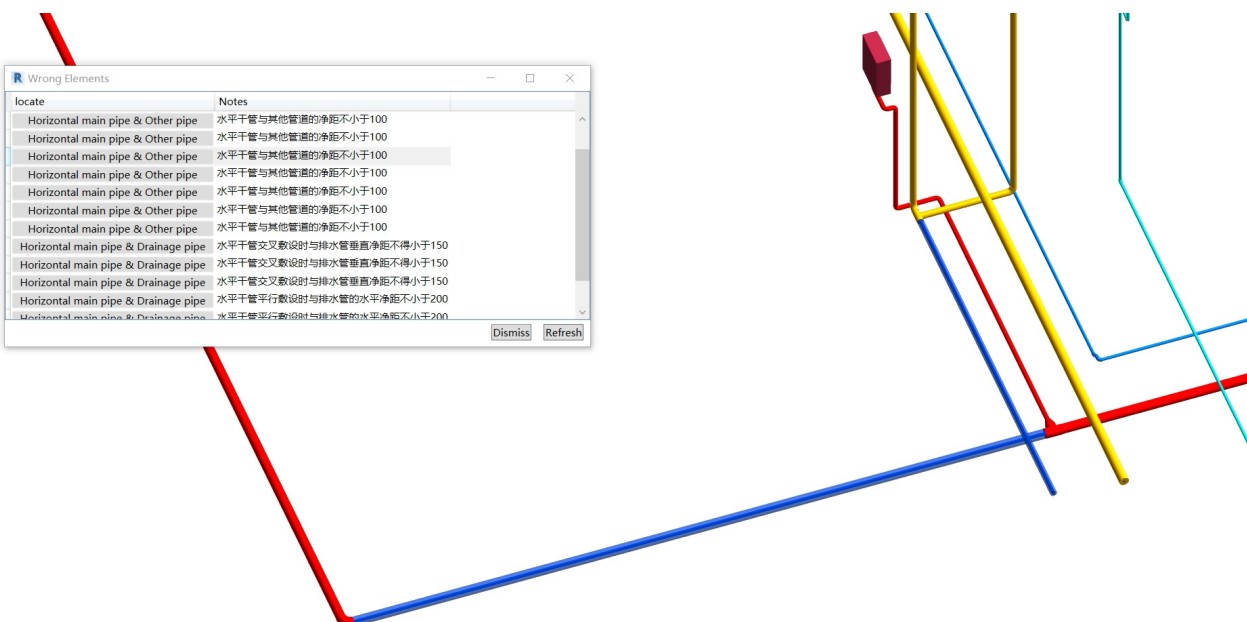

**Figure 17.** Checking results of the element spacing.

## 6. Discussion and Limitation

### 6.1. Discussion of Results

This study realized automated rule checking for MEP by integrating both the MEP KBMS and the BIM-based MEP rule checking system. MEP code requirements were translated into a computer-readable language and documented in the MEP KBMS. The MEP rule checking system is utilized to check the MEP systems integrity of the BIM model, MEP components' properties, and spacing of MEP elements according to the rules listed in the MEP KBMS. This process assists MEP engineers in making decisions. Three cases were examined to examine the system's effectiveness.

To some extent, the MEP knowledge base may be expanded. In "Property" (described in Section 3.2), which has been placed in the knowledge base as independent tables, it may be readily expanded for diameter, slope, or materials by the MEP KBMS (realized in Section 3.4). For "Property" that does not exist in the MEP knowledge base, a new table needs to be added to store them. Additionally, if the new "Property" is to be used for automated rule checking, the system's corresponding codes must be modified. The "Physical property" (described in Section 3.2) as the minimal structure for rule checking is capable of supporting the majority of rule structures, for example, the logic rule-based approach proposed by Kim et al. (2019) and the semantic approach proposed by Guo et al. (2021) [23,33].

In comparison to previous studies, the findings of this study indicate that the original information type carried by the BIM model cannot satisfy the requirements for certain rule checking. Specifically, the types of parameters stored in the BIM model components are primarily physical attributes and do not typically include the function-based parameters expressed in the standard. In light of this finding, this study proposed two methods to increase the model information: individual labeling and centralized labeling. It is relevant to note that this study did not only check the properties of components, but also proposed two algorithms for verifying the spatial relationship between components. As a result, this study extends the range of rule checking that can be applied.

### 6.2. Limitations

However, there are also some limitations to this study. To play a better role, the MEP knowledge base should store rich knowledge. Due to the enormous number of relevant codes and regulations as well as their various expressions, considerable manual efforts are required to convert textual rules into computer-processable language. Meanwhile, it is

challenging to extract rules automatically, so a vast amount of MEP knowledge cannot be collected or optimized. Recent studies on information and communication technologies, such as machine learning [34] and text mining, can provide possible solutions.

In addition, the proposed system aims to provide users with a reliable and efficient tool to perform automated checks. It does not allow users to label the element incorrectly or use an incomplete reliable model, which, however, is inevitable in the use of the system. Currently, preparing the model to be examined needs extensive domain expertise, which may be arduous and time-consuming. The problem is that the system could not assess the annotation after the model with missing information (discussed in Section 4.1) was incorrectly labeled. The system can be improved by analyzing errors generated during use to predict the designers' actions [35].

Finally, this research lacks usability feedback. Since the system is still in the research team's internal usage stage, it has not been extensively utilized, so we do not have enough data to conduct a study on how user-friendly the system is and how long the investment return time is. By collaborating closely with end-users, proactive systems could be developed [36]. Ideally, we would conduct surveys to determine the effectiveness and usability of the current system, and a focus group that uses the system and reports feedback would be required to help evaluate the system.

## 7. Conclusions

This research developed a BIM-based rule checking system for MEP systems compliance checking. For the purpose of knowledge management and programming, the extracted codes are stored in the MEP knowledge base. Additionally, a MEP KBMS was established in order to improve the efficiency of user queries and utilization of codes. This KBMS allows users to edit or delete inappropriate codes, as well as add corresponding codes in line with the actual project situation and best practices. A BIM-based compliance checking system and a KBMS are integrated in this paper. Three case studies verified the validity of this research. The results of this study can simplify the comprehensive review process of MEP components, and provide decision support for engineers, to improve the science and rationality of the MEP design coordination scheme, which has certain practical significance for improving the efficiency and quality of engineering construction. At the same time, it promotes the information construction of normative implementation supervision, and provides potential values for the promotion and application of BIM technology in China.

The suggested future work may focus on the following aspects.

- Automatic rule interpretation. Methods for automatic or semi-automatic extraction of rules using natural language processing (NLP) and ontology are being explored [37–40]. However, due to the complex structure of the regulation provisions [23,33], there is still a long way to go in the research of automatic extraction of regulations.
- Development of BIM standards. The basis of automated compliance checking is to take the objects in the BIM model and compare them with the conceptual objects described in regulations. So only the standardization of the model objects can effectively implement the automated checking of the codes corresponding to them. New requirements are put forward for the development of BIM standards.
- Complex spatial relationship analysis and large-scale reasoning. The current research on automated design review is mostly limited to the examination of component attributes and parameters, and is still insufficient in the analysis and examination of complex spatial relationships. In addition, the existing studies are all based on small-scale data validation, and the research on reasoning and checking in large-scale, complex rules, and spatial relationship scenarios is still blank. It is therefore difficult to meet the actual large and complex engineering design review needs. Breaking through the reasoning and simulation of large-scale, complex rules, and spatial scenarios is one of the key difficulties for the future design review to enter into practicality, automation, as well as comprehensive coverage.

**Author Contributions:** Conceptualization, J.Z. and H.Z.; methodology, X.X.; software, X.X., X.F. and H.Z.; validation, X.X., X.F., R.Z. and Q.B.; formal analysis, X.X., X.F. and R.Z.; investigation, X.X., X.F., R.Z. and Q.B.; resources, R.Z. and Q.B.; data curation, X.F. and R.Z.; writing—original draft preparation, X.X., X.F. and R.Z.; writing—review and editing, J.Z., X.X., X.F. and R.Z.; visualization, X.X. and X.F.; supervision, J.Z.; project administration, X.X. All authors have read and agreed to the published version of the manuscript.

**Funding:** This work was supported by the National Natural Science Foundation of China (Grant number 72171224, 71871116), and The Humanities and Social Sciences Foundation of China's Education Ministry (Grant number 19YJAZH122). The authors would also like to thank the editors and reviewers for their valuable suggestions.

**Institutional Review Board Statement:** Not applicable.

**Informed Consent Statement:** Not applicable.

**Data Availability Statement:** Not applicable.

**Conflicts of Interest:** The authors declare no conflict of interest.

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
