# Peer review of "Automated Rule Checking for MEP Systems Based on BIM and KBMS"

_buildings, doi:10.3390/buildings12070934_

Round 1

Reviewer 1 Report

Introduction

The article, especially the abstract and introduction, requires an extensive editing of English language. There are several grammatical errors which in some cases make it difficult to understand the text. I also suggest avoiding claims such as "It helps to enhance designers’ willingness and capacity to use BIM and improve the quality of engineering construction.” unless proven in a user-centric experiment which is not the case in this study.

Line 43: “Therefore, a method is urgently needed to solve these problems efficiently”. I recommend that authors avoid using sentences like this one which implies that there is a singular unique solution to all of the problems stated. It implies that the proposed work in this paper has the capacity to resolve every issue in a 2D-based design approach which is practically infeasible for any single research paper.

Lines 50-69 could use some citations to support claims regarding the prevailing BIM-based application regarding MEP systems. In addition, the term “volumes of buildings” is ambiguous.

Line 70 defines a generic concept which requires citation.

Line 78: “However prior studies did not establish a complete knowledge base management system (KBMS).” This is a key argument as it outlines a research gap that this study is aiming to address. As such, it requires adequate elaboration of what a “Complete KBMS” means. And in what sense did the previous studies not satisfy such a requirement. Did they ignore a set of rules? If so, is this research going to include every rule related to MEP design and therefore achieve a “Complete KBMS”?

Line 80 is grammatically incorrect. Further the paragraph is long and can be broken into smaller ones.

It appears that the authors are proposing a break-down of dominant domains for BIM-based automated code checking applications which includes Construction and Design processes and argue that the MEP domain did not receive adequate attention. Further the paper claims that the studies were mostly IFC-based which is inefficient because: 1) “IFC-dependent rule checking methods do not take full advantage of relevant knowledge.” And 2) “ Rule checking with the model checker may be inefficient due to the difficulty of data interoperability”.

I do agree that it is generally difficult to work with IFC-based solutions given the challenges of its EXPRESS data schema, but the whole idea behind IFC is open-BIM and interoperability. The second clause is hardly acceptable. Especially given that the solution proposed by the authors relies on a plug-in developed for a certain software. This definitely limits interoperability to a great extent. Not every company uses this software, and many companies use multiple software for their BIM related projects. How can this plug-in be used in such scenarios? Further, even if we assume a project uses this single software, what happens if the versions change? In most cases challenging and expensive software maintenance is needed to keep the plug-in working for every version. Older versions of Revit for instance, do not even support models developed using the newer versions. These sorts of solutions are extremely software dependent which is the primary reason many studies resort to IFC-based solutions. Even the first clause is ambiguous. How does the use of IFC prevent taking advantage of relevant knowledge? For all we know, we can develop ad-hoc solutions (just like the ones in this study) to link IFC knowledge to whatever other source of information we might need. There is an abundance of studies particularly on this subject.

Generally, the introduction lacks major components which are essential for any study:

1-      There is a lack of discussion on what is the current state-of-the-art and what is the gap. The claimed issues in the introduction are either general or inaccurate.

2-       I believe I counted not more than 3 cited articles published after 2019. The article needs a better coverage of recent literature and a more concise discussion of what it is that this particular study is trying to resolve that has not been addressed before.

However, I do find the article contributive in the domain of MEP-based code checking, given that the subject has not received adequate attention compared to other domains. I think a better way to discuss the contribution of this study would be to construct the arguments around this gap and then build the discussion to explain the possible appropriate method for addressing the gap. For instance, a KBMS system could possibly help for these reasons (and then explain the reasons). It’s important to acknowledge that whatever solution that the article proposes, is simply an exploration of an idea not a definite solution to every possible problem in the domain (such an argument can be seen in the third paragraph for instance).

It is also important to acknowledge the previous studies and not to consider them ineffectual (for instance IFC-based studies). The article does not need to prove that it is “better” than an IFC-based solution. Rather it can explain why in this specific case, it is considering the use of a software-dependent solution instead of an open-source one. It is important to list the possible advantages of such an approach, but it is also crucial to discuss possible limitation (such as those I mentioned above). The argument of whether a Revit plug-in is better than an IFC-based solution requires its own research with specific experiments designed to investigate such a matter.

Methodology

Implementation details are well-discussed. Some suggestions are provided below:

Line 129: what does suitable rules mean? This is apparently discussed in line 191. It would be nice to add “ (discussed in section 3.1)” after the term to avoid confusing the readers. I believe there are some studies that defined rules valuable for automation in previous works that can be cited here.

Line 184: This is an important definition and should be provided earlier before discussing the components of the proposed system.

Figure 10 can be improved to fit the texts into shapes. Also what is “ele”? is this defined? I might have missed the definition, but if not provided please add the definition.

Figure 11 has a similar issue of text not fitting the shape.

Case study

I suggest changing the word ‘window’ to ‘user interface’ in sentences such as Figure 13 caption.

The design of the case study is not for evaluation and is rather a demonstration on how to use the proposed system.

Evaluation requires a detailed design of experiment, an expected outcome, and an analysis of the actual outcome all of which are absent in the current study. For instance, does the system capture all non-compliant components? Right now, a single non-compliant example is presented, but it is not clear whether these examples were the only ones.

Further, evaluation has aspects beyond technical matters. It should also include an analysis of how user friendly the system is and whether the investments required to develop the proposed system has a reasonable return on investment. Such matters can only be determined in real-life applications performed by regular designers in controlled experiments. For instance, a focus group that uses the system and reports feedback.

Understandably, tackling all of the aspects in an evaluation process is difficult. But it is essential to acknowledge the need for such evaluations and at least discuss this as parts of the limitations of the study.

Discussion and conclusion

The discussion initiates with a review of the literature that should have been provided in earlier sections before the methodology. The first two paragraphs should be used to justify the research gap in the beginning of the study.

Once again, I do not agree with the claim of the article that it has successfully addressed the proposed research gap. For instance:

Line 501: “But it is still a huge problem to carry out an effective automatic MEP assessment using BIM technology while taking full use of BIM’s technological benefits”.  The claim that all previous studies had this issue and the issue is now resolved in this one is a bit difficult to agree with.

Some discussions need to be added:

1-      An important limitation is the structure of rules that can be supported with the proposed rule presentation system. Can the system be extended to cover other rule structures or we will need fundamentally different systems?

2-      What would be the sensitivity of the system to change and scalability? For instance, if we decided to add another table to the relational schema in Figure 5, how does it affect the system? Will we need to redevelop the BIM-based MEP system?

3-      What if someone uses the proposed system on models that are labeled incorrectly? Do we expect the designers to always use the right labels? If so, this should be included in the limitations of the approach.

Finally, the paper is overly focused on technical matters. There are non-technical limitations when it comes to code checking in general. In practice, there are several challenges that a designer has to deal with in order to implement a code checking system. Some examples are the need for a complete reliable model or using standardized labeling systems which makes the process subject to error.

 I strongly suggest having a look at one of the recent articles on automated rule checking best practices to improve the discussions regarding the research gap and limitations.

Reviewer 2 Report

I congratulate the authors for this article, which offers an interesting contribution to the BIM methodology. The text provides sufficient references and is written in an orderly and coherent manner. On a minor note, it is increasingly shocking to refer to BIM as an emerging methodology, as it has been under development for years.

As for the specific content of the article, although it deals with the basic checking rules (integrity, properties, spacing), it is an adequate starting point, which can be extended to other checks. The proposed future developments are promising.

Author Response

Thank you for your comments concerning our manuscript entitled “Automatic rule checking for MEP system based on BIM and KBMS” (ID: buildings-1763873). These comments are all valuable and very helpful in revising and improving the paper, as well as guiding us in our research.

We are sorry for the confusion that we refer BIM as an emerging method. What we're trying to express is that BIM has been improving steadily in recent years. In the revised manuscript, we have changed this in section 1.  

Thank you again for your positive comments and valuable suggestions to improve the quality of our manuscript. We also believe that in the future, automated rule checking for MEP system based on BIM would flourish.

Reviewer 3 Report

Dear authors,

Thanks for your contribution to Sustainability.

Before further process of this manuscript, please check if it matches the scope of the journal.

With major revisions of the manuscript, it might be reconsidered.

The opinions are set out below:

STRUCTURE 

Please prepare the manuscript following the instructions for authors. 

ENGLISH

The manuscript has several typos. Authors need to proofread the paper to eliminate all of them.

Some sentences are too long. Generally, it is preferable to write short sentences with one idea in each sentence.

REFERENCES

The literature review is incomplete. Several relevant references are missing. The reference list should include the full title, as recommended by the style guide.

INTRODUCTION 

Authors should include additional references in the introduction that support the claims. 

Authors should better explain the background to this research, including why the research issue is important. Contributions should be enhanced. It should be made clear what is novel and how it addresses the limitations of prior work.

RELATED WORK 

The related work section is not well organized. Writers should try to categorize articles and present them logically. Authors should add a table comparing the main features of previous work in order to highlight their differences and limitations. Alternatively, authors may consider adding a row to the table to describe the proposed solution.

PROBLEM DEFINITION 

Authors should provide a clear and detailed definition of the issue. Authors should include an example to illustrate how the problem is defined.

METHOD

A novel solution is presented, but it is important to better explain the design decisions (e.g. why the solution is designed that way). There is a need for discussion of the complexity of the proposed solution.

CASE STUDY

Case studies should be updated to incorporate some comparisons with more recent studies.

Sincerely yours,

Round 2

Reviewer 1 Report

Overall, the quality of the paper is greatly improved. There are some minor suggestions that authors should consider:

1-     Almost half of the discussion is regarding the limitations of the work. I suggest adding a subsection (6.1 limitations) and put the text under this topic.

2-     The text that was added in the revision comes with writing issues. Kindly use freely available tools such as Grammarly to check on these. Some examples:

a.      Line 538: “… a uncomplete …” should be changed to “ … an incomplete … “

b.      I believe the sentence in line 540 should be in present tense: “The problem is … “

c.      Line 544: “Finally, this research lakes …” .  should be “lacks”.

3-     It is good practice to cite possible relevant papers that has addressed your limitations to some extent. For instance, in line 533, ‘Recent studies on information and communication technologies …’. This could benefit from a citation to such a study. The second and third limitations are also examined in some studies. For instance, the below reference is a good example of how a rule checker system affects the daily design processes:

·        Lee, H.W., Oh, H., Kim, Y. and Choi, K., 2015. Quantitative analysis of warnings in building information modeling (BIM). Automation in Construction51, pp.23-31.

The below study focuses on the non-technical processes and how we need evolutionary and proactive approaches in designing rule checkers:

·        Sobhkhiz, S., Zhou, Y.C., Lin, J.R. and El-Diraby, T.E., 2021. Framing and Evaluating the Best Practices of IFC-Based Automated Rule Checking: A Case Study. Buildings11(10), p.456.

The below study is also useful for the investment analysis of BIM-based systems:

·        Oesterreich, T. D., & Teuteberg, F. (2018). Looking at the big picture of IS investment appraisal through the lens of systems theory: A System Dynamics approach for understanding the economic impact of BIM. Computers in Industry, 99, 262-281.

Lastly, I want to congratulate the authors for their hard work.

Author Response

Thank you for your comments concerning our manuscript entitled “Automatic rule checking for MEP system based on BIM and KBMS”.

1-     Almost half of the discussion is regarding the limitations of the work. I suggest adding a subsection (6.1 limitations) and put the text under this topic.

Response:

Thank you for your constructive suggestion. In the revised manuscript, the discussion has been divided into two sections (6.1 Discussion of Results and 6.2 Limitations).

2-     The text that was added in the revision comes with writing issues. Kindly use freely available tools such as Grammarly to check on these. Some examples:

  1. Line 538: “… a uncomplete …” should be changed to “ … an incomplete … “
  2. I believe the sentence in line 540 should be in present tense: “The problem is … “
  3. Line 544: “Finally, this research lakes …” . should be “lacks”.

Response:

Thanks very much. Issues have been corrected in this version.

3-     It is good practice to cite possible relevant papers that has addressed your limitations to some extent. For instance, in line 533, ‘Recent studies on information and communication technologies …’. This could benefit from a citation to such a study. The second and third limitations are also examined in some studies. For instance, the below reference is a good example of how a rule checker system affects the daily design processes:

  • Lee, H.W., Oh, H., Kim, Y. and Choi, K., 2015. Quantitative analysis of warnings in building information modeling (BIM). Automation in Construction, 51, pp.23-31.

The below study focuses on the non-technical processes and how we need evolutionary and proactive approaches in designing rule checkers:

  • Sobhkhiz, S., Zhou, Y.C., Lin, J.R. and El-Diraby, T.E., 2021. Framing and Evaluating the Best Practices of IFC-Based Automated Rule Checking: A Case Study. Buildings, 11(10), p.456.

The below study is also useful for the investment analysis of BIM-based systems:

  • Oesterreich, T. D., & Teuteberg, F. (2018). Looking at the big picture of IS investment appraisal through the lens of systems theory: A System Dynamics approach for understanding the economic impact of BIM. Computers in Industry, 99, 262-281.

Response:

We sincerely appreciate the valuable comments. We have checked the literature carefully and added these references to the discussion part.

Finally, we would like to express our sincere thanks to you again.

Reviewer 3 Report

Accept

Author Response

Thanks again for your comments concerning our manuscript entitled “Automatic rule checking for MEP system based on BIM and KBMS”.